# Dual ubiquitin signalling by SLOMO controls AUX1 activity and turnover during root gravitropism

Lixia Pan[1,2,9], Shanshuo Zhu [1,2,3,4,8,9], Shao-Li Yang [1,2], Nathan Mellor[5], Francesca R Iacobini[6], Tingyu Zhu [1,2], Pia Neyt[1,2], Brigitte van de Cotte[1,2], Michaël Vandorpe[1,2], Ranjan Swarup[5], Daniël Van Damme[1,2], Markus M Geisler [6], Kris Gevaert[3,4], Leah R Band[5,7] & Ive De Smet [1,2]✉

## Abstract

Gravity-directed growth ensures that shoots and roots grow upwards and downwards, respectively. To achieve this, the organ's angle with respect to gravity drives the asymmetric redistribution of the plant hormone auxin and consequently differential elongation, thus creating a curvature of the organ. In addition to efflux transporters, the auxin influx transporter AUXIN RESISTANT 1 (AUX1) is essential for auxin redistribution during root gravitropism. Here, we show that the F-box protein SLOMO regulates AUX1 via two distinct mechanisms. First, SLOMO promotes proteolytic degradation of AUX1, possibly in an indirect manner. Second, SLOMO controls the ubiquitination of K261, K264, and K266, thus potentially affecting AUX1 auxin transport properties that are regulated by these residues. This reveals a dual mode of SLOMO-mediated regulation of AUX1, including a novel, non-proteolytic role for SLOMO-mediated ubiquitination in addition to SLOMO-mediated degradation of AUX1.

**Keywords** Auxin; Gravitropism; Root; F-box Protein; Ubiquitination
**Subject Categories** Plant Biology; Post-translational Modifications & Proteolysis

## Introduction

Plant growth and development are affected by environmental signals, which can be variable (e.g., heat or drought) or constant (e.g., gravity). Gravity-directed growth (or gravitropism), ensures that shoots and roots typically grow upwards and downwards, respectively. To achieve this, the organ's angle with respect to gravity drives the asymmetric redistribution of the plant hormone auxin and consequently differential elongation, creating a curvature of the organ (Sack and Kiss, 1989; Friml et al, 2002; Luschnig et al, 1998; Blancaflor and Masson, 2003; Swarup et al, 2005). This auxin redistribution requires regulation of auxin influx and efflux carriers at multiple levels, including transcription, polarized subcellular protein localization, protein modification, and protein recycling and degradation, with several of these processes controlled by post-translational modifications (Zazimalova et al, 2010; Muday and Murphy, 2002; Adamowski and Friml, 2015; Vanneste et al, 2025). Following gravistimulation, more auxin is directed into the elongation zone at the lower side of the root when compared to the upper side, generating an auxin gradient that causes inhibition of growth on the lower side of the root and thus making the root bend (Luschnig et al, 1998; Müller et al, 1998; Bennett et al, 1996; Ottenschläger et al, 2003; Young et al, 1990; Vanneste et al, 2025). In the Arabidopsis root, the auxin influx transporter AUXIN RESISTANT 1 (AUX1) and the efflux transporters PIN-FORMED 2 (PIN2) and ATP-BINDING CASSETTE B1 (ABCB1) play a crucial role in auxin redistribution to mediate root gravitropism (Swarup and Bhosale, 2019; Abas et al, 2006; Baster et al, 2012; Wang et al, 2013; Shan et al, 2011; Muday and Rahman, 2007). The reduction of PIN2 at the upper side is due to PIN2 ubiquitination and subsequent vacuolar targeting of PIN2 (Kleine-Vehn et al, 2008; Leitner et al, 2012). Specifically, lysine63-linked ubiquitination mediates PIN2 sorting to the multivesicular body by the endosomal sorting complex required for transport for subsequent vacuolar degradation, which is important for auxin distribution in root meristems and further affects root growth during environmental adaptation (Kleine-Vehn et al, 2008; Leitner et al, 2012). While AUX1 is essential within lateral root cap and elongating epidermal cells for gravitropic response (Swarup et al, 2005), little is known about the post-transcriptional regulation of AUX1 in the control of root gravitropism (Swarup and Bhosale, 2019; Konstantinova et al, 2021). Overall, the AUX1 structure adopts an inward-facing conformation, and in the auxin-bound structure, indole-3-acetic acid (IAA) is coordinated to AUX1 primarily through hydrogen bonds with its carboxyl group (Yang et al, 2025).

Ubiquitination is a highly regulated process that marks proteins for degradation via the proteasome or, in the case of integral membrane proteins, via clathrin-mediated endocytosis, alters their cellular location, affects their activity, and promotes or prevents interactions with other proteins in eukaryotic cells (Glickman and

[1]Department of Plant Biotechnology and Bioinformatics, Ghent University, Ghent, Belgium. [2]VIB Center for Plant Systems Biology, Ghent, Belgium. [3]VIB-UGent Center for Medical Biotechnology, VIB, Ghent, Belgium. [4]Department of Biomolecular Medicine, Ghent University, Ghent, Belgium. [5]Division of Plant and Crop Sciences, School of Biosciences, University of Nottingham, Nottingham, UK. [6]Department of Biology, University of Fribourg, Fribourg, Switzerland. [7]Centre for Mathematical Medicine and Biology, School of Mathematical Sciences, University of Nottingham, Nottingham, UK. [8]Present address: Faculty of Biology & Biotechnology, Ruhr-University Bochum, Bochum, Germany. [9]These authors contributed equally: Lixia Pan, Shanshuo Zhu. ✉E-mail: ive.desmet@psb.vib-ugent.be

Ciechanover, 2002; Mukhopadhyay and Riezman, 2007; Schnell and Hicke, 2003; De Meyer et al, 2023). Ubiquitination of lysine residues involves an enzymatic cascade with three main steps, namely ubiquitin activation, conjugation, and ligation, performed by ubiquitin-activating enzymes (E1s), ubiquitin-conjugating enzymes (E2s), and ubiquitin ligases (E3s), respectively (Swatek and Komander, 2016; Ho et al, 2006). E3 ubiquitin ligases have been shown to regulate the directional transport of auxin, exemplified by the RING-finger E3 ligase WAVY GROWTH 3 (WAV3) (Konstantinova et al, 2021). Among the E3 ligases, F-box proteins are part of the Skp-Cullin-F-box (SCF) complex and, in Arabidopsis, there are around 700 F-box proteins (Xu et al, 2009; Ho et al, 2006). F-box proteins play vital roles in plant development, responding to biotic/abiotic stresses, and phytohormone perceptions (Thines et al, 2007; Dharmasiri et al, 2005; Kelley and Estelle, 2012; Rao and Virupapuram, 2021; Zhang et al, 2019). One of these F-box proteins, SLOW MOTION (SLOMO), is involved in organ initiation and hypocotyl growth (Lohmann et al, 2010; Zhu et al, 2024). While it was proposed that SLOMO is required for normal auxin distribution in the plant, the underlying mechanism is largely unknown (Lohmann et al, 2010). Here, we used a novel role of SLOMO in gravitropism to unravel its role in ubiquitination-mediated control of AUX1 levels and activity.

## Results and discussion

To gain deeper insight into the roles and targets of the hardly explored F-box protein SLOMO, we revisited *slomo* loss-of-function phenotypes. In addition to other typical auxin-mediated phenotypes (Appendix Fig. S1) (Zhu et al, 2024; Lohmann et al, 2010), we observed a faster and exaggerated primary root response to gravity (gravitropism) of *slomo* mutants, compared to Col-0 (Fig. 1A–C; Appendix Fig. S2). In contrast, 35S-mediated over-expression of *mCherry:SLOMO* (*35S::mCherry:SLOMO*) (Appendix Fig. S3a,b) resulted in a delayed primary root response to gravity compared to Col-0 (Fig. 1D,E; Appendix Fig. S3c). Root gravitropic bending is accompanied by asymmetric auxin distribution and response with the maximum at the lower side of gravity-stimulated roots, which can be monitored with the transcriptional auxin response reporter *pDR5::GUS* (Sabatini et al, 1999; Buer and Muday, 2004; Band et al, 2012; Konstantinova et al, 2021). To assess auxin distribution and response, vertically grown Col-0 and *slomo-3* seedlings expressing *pDR5::GUS* were subjected to a 90° gravity stimulus, and the expression pattern of *pDR5::GUS* was examined over time. The typical increased GUS activity at the lower side of the root was observed significantly earlier in *slomo-3* compared to Col-0 (Fig. 1F). Since the above-mentioned phenotypes are associated with auxin transport and in agreement with a genetic interaction of SLOMO and components of polar auxin transport (Konstantinova et al, 2021; Lohmann et al, 2010), we analyzed the effect of an auxin efflux inhibitor (NPA) and an auxin influx inhibitor (2-NOA) on primary root length. The response of *slomo-3* roots to NPA treatment was similar to Col-0 (Fig. 1G). In contrast, *slomo-3* displayed a higher reduction in primary root length upon 2-NOA treatment than Col-0 (Fig. 1G). The auxin transporters AUX1 (auxin influx carrier) and PIN2 (auxin efflux carrier) are required for the asymmetric distribution of auxin driving the root gravitropic response (Marchant et al, 1999;

Rahman et al, 2010; Swarup and Bhosale, 2019). The higher sensitivity of *slomo-3* roots to 2-NOA suggested that auxin influx rather than auxin efflux is altered in *slomo-3*. This is further supported by an increased sensitivity of *slomo-3* to the synthetic auxin 2,4-dichlorophenoxyacetic acid (2,4-D), the uptake of which is facilitated by AUX1 (Hoyerova et al, 2018; Marchant et al, 1999) (Fig. 1H). To confirm this further, we measured auxin influx in Col-0 compared to *slomo-3*, and this revealed increased auxin influx in *slomo-3* (Fig. 1I).

We next showed that *SLOMO* is expressed in the root tip, with strong expression in the root cap and root epidermis, which overlaps with the cells where the AUX1 protein is found (Swarup and Péret, 2012) (Appendix Fig. S4). We did not observe any differential expression of *SLOMO* between the upper and the lower side of the root tip upon gravistimulation in the *pSLOMO::NLS:GFP* line (Appendix Fig. S5a–c). A similar analysis of a *pSLOMO:::GFP:SLOMO* line showed a very small shift in the lower side/upper side ratio of the (lowly abundant) GFP:SLOMO protein in the root tip upon gravistimulation (Appendix Fig. S5d–f). Analyses of a *slomo-3 aux1-22* double mutant, which exhibited a phenotype like that of the *aux1* single mutant (Appendix Fig. S6), suggested that the *slomo-3* phenotype requires AUX1 activity. Since AUX1 drives gravitropic responses from the lateral root cap and/or epidermis (Swarup et al, 2005), we explored if this overlapped with SLOMO activity. Indeed, expressing *SLOMO* in the lateral root cap and/or epidermis using a tissue-specific promoter (Wendrich et al, 2020) in the *slomo-3* mutant background rescued the *slomo-3* phenotype (Appendix Fig. S7).

We subsequently explored whether the F-box protein SLOMO interacts with AUX1. To stabilize the interaction with putative E3-targets, we used a SLOMO-decoy protein variant that lacks the F-box domain and is still able to bind to substrates, but lacks the ability to be recruited into CULLIN 1 ligase complexes to mediate substrate ubiquitination and degradation (Feke et al, 2019; Zhu et al, 2024) (Appendix Fig. S8). Using co-immunoprecipitation (co-IP) following transient co-expression of *mCherry:SLOMO-Decoy* with *AUX1:YFP* in *Nicotiana benthamiana*, we revealed an interaction between AUX1:YFP and mCherry:SLOMO-Decoy (Fig. 2A). This interaction can be direct or indirect, as co-IP experiments do not allow to discriminate between these two possibilities. The association of SLOMO and AUX1 was further confirmed by a co-IP assay using transgenic *aux1-22* plants carrying YFP-tagged AUX1 expressed under the control of its endogenous promoter (*pAUX1::AUX1:YFP*) and mCherry-tagged SLOMO expressed under the control of the 35S promoter (*35S::mCherry:SLOMO*) (Fig. 2B; Appendix Fig. S9) and through a split-ubiquitin yeast assay, although the interaction could only be observed in very few independent transformants (Appendix Fig. S10). The low frequency of yeast transformations showing interaction might indicate a need for specific expression levels of the proteins, targeting, for example, sufficient amounts to the proper location, to yield interaction in this system. AUX1 belongs to a small gene family comprising of four highly conserved genes, *AUX1* and *LIKE-AUX1* (*LAX*) genes, *LAX1*, *LAX2*, and *LAX3*, which originated from a common ancestor through the process of gene duplication with well-conserved gene structure (Péret et al, 2012). Using co-IP following transient co-expression of *mCherry:SLOMO-Decoy* with *LAX1:VENUS* and *LAX2:VENUS* in *Nicotiana benthamiana*, we revealed that mCherry:SLOMO-Decoy

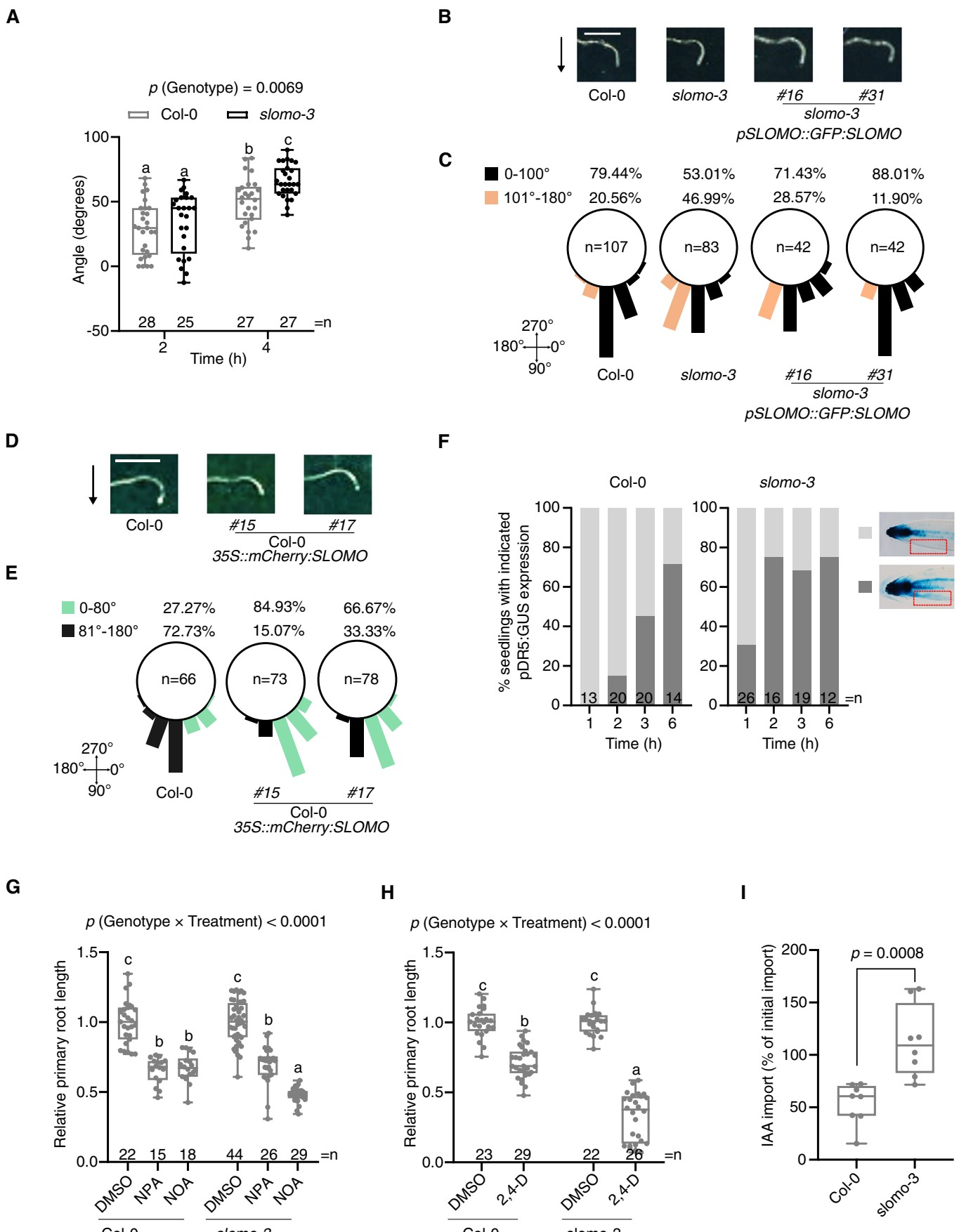

**Figure 1.   SLOMO plays a role in root gravitropism.**

(A) Quantification of the gravitropic bending angle of Col-0 and *slomo-3* over time following gravistimulation. Box plots with individual data points represent the distribution of all individual root bending angles. Letters indicate significant differences based on two-way ANOVA. The exact *p*(Genotype) value between Col-0 and *slomo-3* is shown at the top. (B, E) Representative images (B, D) and distribution of root bending angles (C, E) after 6 h of gravistimulation. Scale bar = 2 mm (B, D). Arrows indicate the direction of the gravity vector (B, D). Percentage of root bending angle at 20° intervals. The number of individual seedlings analyzed for each line (*n*) is shown within the corresponding circles. The 0° starting position upon gravistimulation and major directions after bending (90°, 180°, and 270°) are indicated. For (C), the root bending angle is less than 100° in black, and more than 100° in orange, and the percentage is shown above the circle. For (E), the root bending angle is less than 80° in green, and more than 80° in black, and the percentage is shown above the circle. (F) Quantification of pDR5::GUS pattern in Col-0 and *slomo-3*. The percentage of *DR5* expression with two representative patterns at each time point is shown. The number of individual seedlings (*n*) for GUS staining indicated above the *X* axis. (G) Relative total primary root length (within a genotype) of Col-0 and *slomo-3* on 5 µM NPA and 1 µM 2-NOA. (H) Relative total primary root length (within a genotype) of Col-0 and *slomo-3* on 50 nM 2,4-D. Box plots (G, H) with whiskers down to the minimum and up to the maximum value, and each individual value as a dot superimposed on the graph. The number of individually measured seedlings (*n*) is indicated above the *X* axis. The respective letters indicate significant differences based on two-way ANOVA with Tukey's HSD ($P < 0.05$). The *P* value for the interaction (Genotype × Treatment) is shown at the top. (I) Quantification of indole-3-acetic acid (IAA) import in mesophyll protoplasts from Col-0 and *slomo-3* seedlings. Box plots with individual data points represent the distribution of IAA import across seven biological replicates. The *P* value comparing genotypes is shown at the top. For all box plot panels (A, G, H, I), the box spans from the 25th percentile to the 75th percentile of the data. A horizontal line inside the box marks the median (50th percentile). Whiskers extend from the minimum to the maximum. For (B–E), two biological replicates were performed with similar results. Source data are available online for this figure.

also interacts with LAX1 and LAX2 (Fig. 2A). The observation that SLOMO can interact with several members of the AUX1/LAX family indicates a broader role for SLOMO in AUX1/LAX-mediated processes.

F-box proteins, such as SLOMO, are responsible for substrate recognition by multi-protein E3 ubiquitin ligase complexes that typically catalyze the ubiquitination of proteins destined for 26S proteasomal degradation (Lechner et al, 2006). Furthermore, ubiquitination can also have non-proteolytic roles, such as membrane trafficking, protein kinase activation, DNA repair, and chromatin dynamics (Chen and Sun, 2009). For example, ubiquitin marks plasma membrane cargo for internalization (Schwihla and Korbei, 2020), and the closely related ubiquitin-conjugating enzymes UBC35 and UBC36 are the main sources of K63-linked ubiquitin chains that mediate vacuolar degradation of plasma membrane proteins in Arabidopsis (Romero-Barrios et al, 2020). Given that SLOMO and AUX1 interact, it is likely that SLOMO ubiquitinates AUX1. To evaluate whether a fraction of AUX1 is post-translationally modified by ubiquitination, immunoprecipitated AUX1:YFP was probed with the general P4D1 anti-ubiquitin antibody that recognizes monoubiquitin and several forms of polyubiquitin chains. We observed a high molecular weight smear (55 kDa—higher), typical of ubiquitinated proteins, in immunoprecipitates from *aux1-22 pAUX1::AUX1:YFP* plants (Fig. 2C,D). In contrast, we observed a much weaker high molecular weight smear in *aux1-22 pAUX1::AUX1:YFP slomo-3* compared to *aux1-22 pAUX1::AUX1:YFP* (Fig. 2C,D). Taken together, this supports that SLOMO ubiquitinates AUX1.

Since SLOMO ubiquitinates AUX1, we explored if the protein level of AUX1 was affected in *slomo* mutants. Indeed, we observed higher AUX1 levels in *slomo* mutant roots compared to Col-0 through Western blot (Fig. 3A,B). Microscopic analyses of *pAUX1::AUX1:YFP* in *slomo-3 aux1-22* compared to *pAUX1::-AUX1:YFP* in *aux1-22* supported the higher AUX1 levels in *slomo-3* (Fig. 3C,D; Appendix Fig. S11). In contrast, transient over-expression of *TagRFP:SLOMO* and *AUX1:YFP* in Arabidopsis protoplasts resulted in lower AUX1:YFP levels compared to overexpression of *AUX1:YFP* alone (Appendix Fig. S12) and overexpression of *mCherry:SLOMO* in Col-0 resulted in lower

AUX1 levels compared to Col-0 (Fig. 3E,F). We confirmed that these differences in AUX1 levels in *slomo* mutants and *SLOMO* overexpression lines are not caused by elevated *AUX1* expression (Appendix Fig. S13). Finally, similar to AUX1 protein accumulation in *slomo-3*, treatment with concanamycin A (ConA), which inhibits vacuolar degradation in Arabidopsis roots (Merkulova et al, 2014) led to accumulation of AUX1 protein in *aux1-22 pAUX1::AUX1:YFP* (Fig. 3G,H). In contrast, treatment with the proteasome inhibitor MG132 (GUO and PENG, 2013) and Bortezomib (Kisselev et al, 2012) did not affect AUX1 protein levels (Fig. 3G,H; Appendix Fig. S14). These results support that SLOMO regulates AUX1 protein levels, possibly through vacuolar targeting. However, we did not observe any striking differences in AUX1:YFP plasma membrane localization comparing *aux1-22 pAUX1::AUX1:YFP* and *aux1-22 pAUX1::AUX1:YFP slomo-3* (Appendix Fig. S15).

Mathematical modeling predicted that globally increasing or decreasing AUX1-mediated auxin flux had a limited effect on auxin redistribution upon a gravitropic stimulus (Fig. 3I,J; Appendix Fig. S16). In contrast, the model predicted that increasing AUX1-mediated auxin flux through increased AUX1 levels (also reflecting increased activity) only on the lower side of the root tip resulted in a faster establishment of the auxin asymmetry, with higher auxin levels on the lower side (Fig. 3I,J). Simulation of other possible scenarios for AUX1 activity between the upper and the lower root sides either disrupted or did not change the speed in establishing the auxin asymmetry (Appendix Fig. S16). Therefore, the change in AUX1 levels downstream of SLOMO does not explain the observed increase in auxin asymmetry and gravitropic response and might be an (indirect) side effect. We therefore hypothesized that (some) SLOMO-mediated ubiquitination sites could have other roles.

Therefore, we next attempted to identify *in planta* ubiquitinated AUX1 residues, but so far, we have not been able to identify these. Therefore, to pinpoint relevant ubiquitination sites for further functional analyses, we focused on highly conserved lysines as ubiquitination target sites in the AUX1 intracellular hydrophilic regions (Fig. 4A; Appendix Fig. S17) and especially in the region that was most associated with agravitropic mutants (Swarup et al, 2004; Singh et al, 2018) (Fig. 4A; Appendix Fig. S17). Next, we

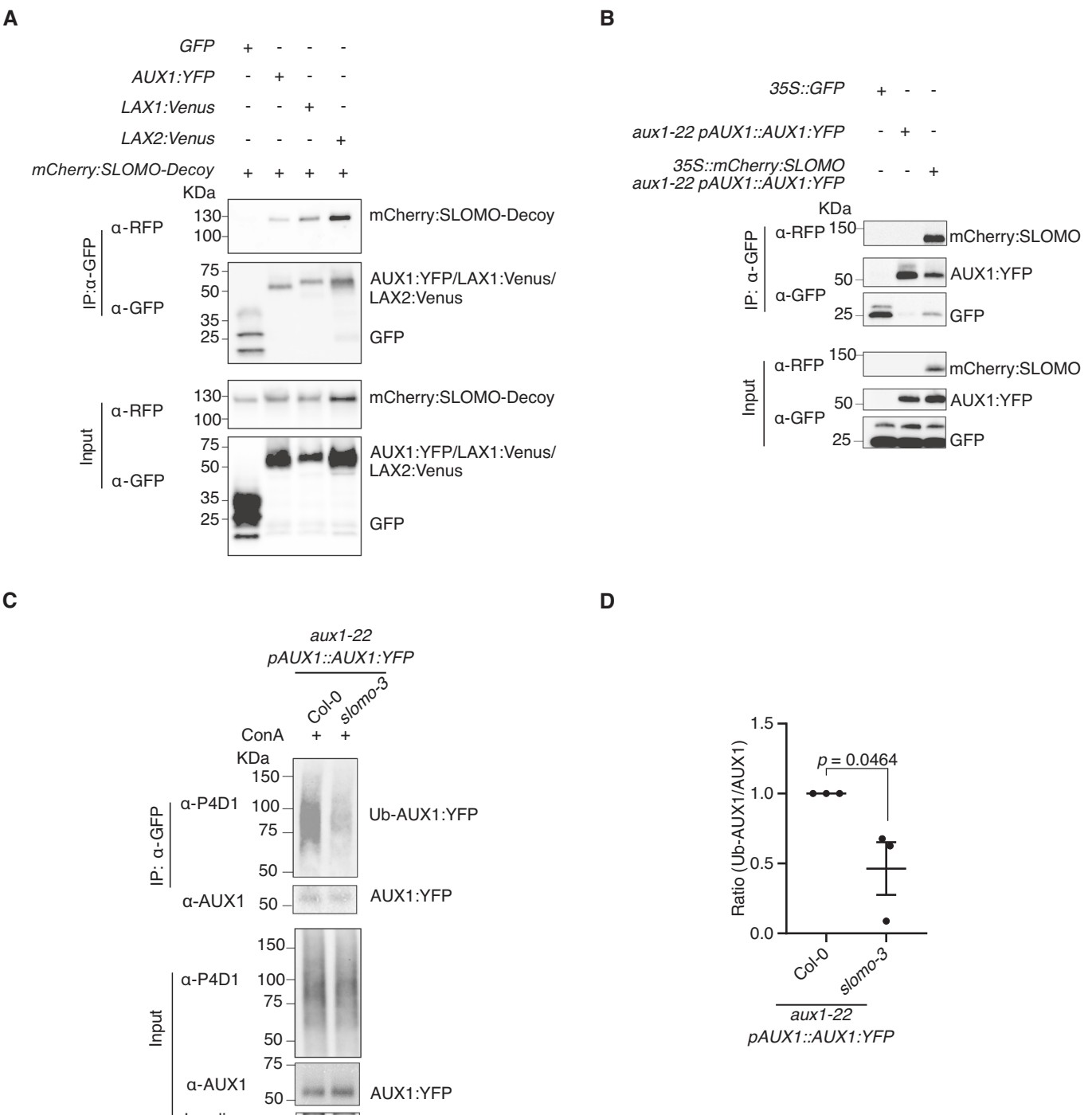

**Figure 2. SLOMO interacts with AUX1 and influences AUX1 ubiquitination.**

(A) Representative Western blot for co-IP between mCherry:SLOMO-Decoy and AUX1:YFP, LAX1:Venus, or LAX2:Venus upon transient expression in *N. benthamiana*. (B) Representative Western blot for co-IP between mCherry:SLOMO and AUX1:YFP in *Arabidopsis*. Input, immunoblots showing the abundance of indicated proteins in the total protein extracts. Immunoprecipitation (IP) products precipitated by the anti-GFP antibody. Total proteins (Input) or IP products were probed with an anti-RFP or anti-GFP antibody. (C, D) Detection (C) and quantification (D) of AUX1 ubiquitination in Col-0 and *slomo-3* background following AUX1:YFP IP. IP products precipitated by the anti-GFP antibody. Ub-AUX1:YFP was detected with α-P4D1 antibody, and AUX1-YFP was detected with α-AUX1 antibody. The bands on the stain-free gel are the loading control (C). (C) Equal loading of AUX1:YFP was used to focus on the difference in ubiquitination. Quantification of the ratio between Ub-AUX1 and AUX1 from three biological replicates (indicated with dots) (D), The graph shows the mean of three biological replicates (individual dots) with standard error of the mean (SEM). Student's *t* test with two-tailed distribution and two-sample equal variance was used to calculate the indicated *P* value. Two (A, B) or three (C) biological replicates were performed with similar results. Source data are available online for this figure.

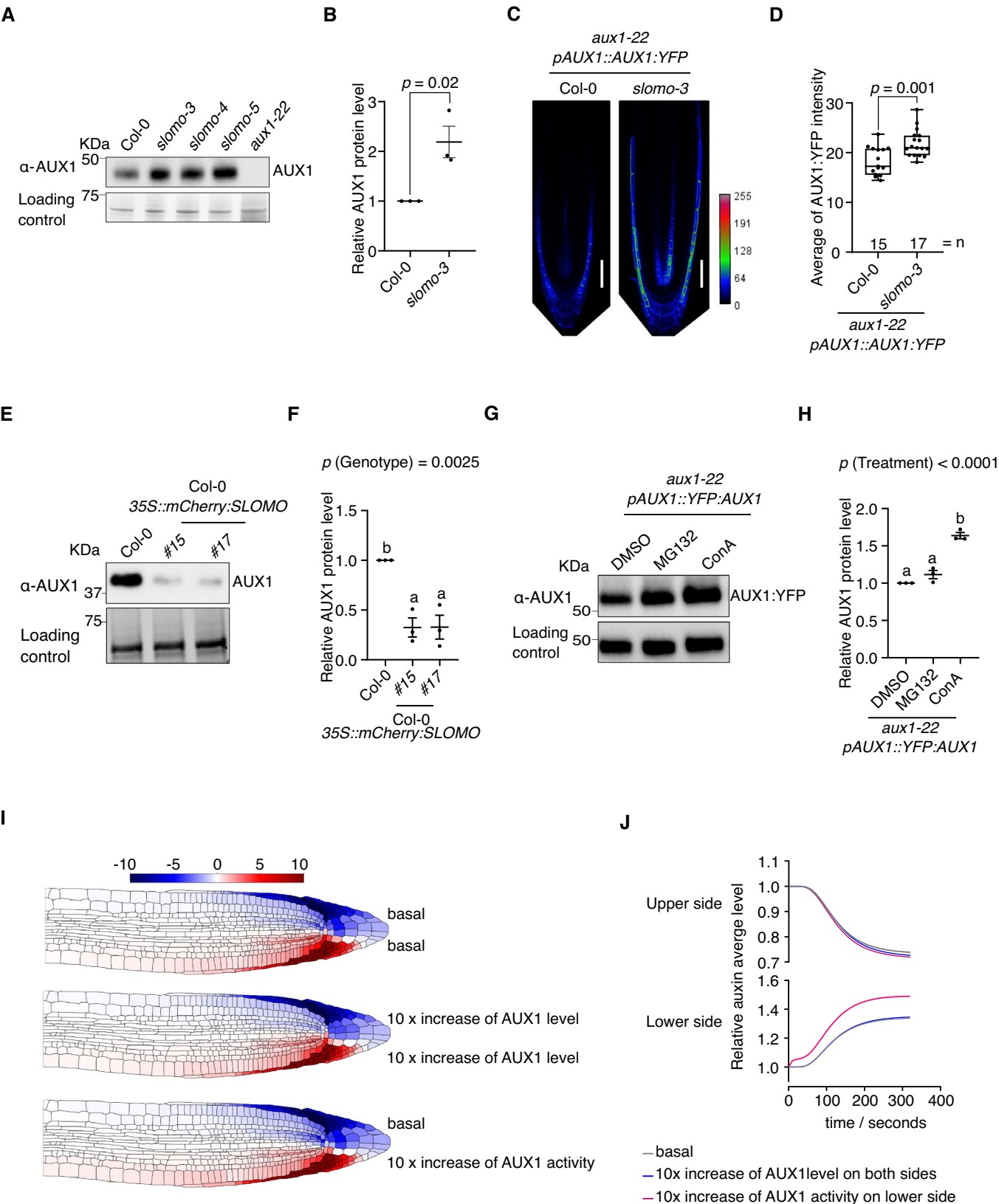

**Figure 3. SLOMO promotes AUX1 degradation.**

(A, B) Representative Western blot showing endogenous AUX1 protein in Col-0 and *slomo* mutants. Bands on stain-free gel as loading control (A). Quantification of the relative level of AUX1 (AUX1/loading control) in *slomo-3* versus Col-0. The graph shows the mean of three biological replicates (individual dots) with standard error of the mean. Student's *t* test with two-tailed distribution and two-sample equal variance was used to calculate the indicated *P* value (B). (C, D) AUX1:YFP signal in wild-type and *slomo-3* background. Representative confocal rainbow images display high or low levels of AUX1:YFP in the primary root tip as indicated through a color gradient. Scale bar = 100 μm (C). Quantification of the average AUX1:YFP fluorescence intensity of up to 17 images with an intensity threshold >4. Box plots with individual data points. The number of analyzed seedlings for each genotype is shown above the *X* axis (*n*). Student's *t* test with two-tailed distribution and two-sample equal variance was used to calculate the indicated *P* value. Individual data points are plotted to ensure full transparency. The box plot center represents the median (50th percentile), while the box bounds indicate the 25th and 75th percentiles. The whiskers extend from the minimum to the maximum values (D). (E, F) Representative western blot detecting endogenous AUX1 protein in Col-0 and *35S::mCherry:SLOMO*. Bands on stain-free gel as loading control (E). Quantification of the relative level of AUX1 (AUX1/loading control). The graph shows the mean of three biological replicates (individual dots) with standard error of the mean (SEM). Different letters denote significant differences (*P* < 0.05) based on one-way ANOVA with Tukey's HSD. The *P* value among genotypes (*P* = 0.0025) is shown at the top of the graph (F). (G, H) Representative Western blot showing AUX1 abundance after 50 μM MG132 and 1 μM ConA treatment. Bands detected by α-Tubulin as a loading control. Quantification of the relative level of AUX1 (AUX1/loading control). The graph shows the mean of three biological replicates (individual dots) with SEM. Different letters denote significant differences (*P* < 0.05) based on one-way ANOVA with Tukey's HSD. The *P* value among genotypes (*P* < 0.0001) is shown at the top of the graph (G). AUX1 detected by α-AUX1 antibody in (A, E, G). (I, J) Model predictions for auxin (re)distribution in the root tip upon a gravitropic stimulus, with basal AUX1 levels, with ten times global increase in AUX1 levels, and with a ten times increase of AUX1 levels in the lower root side. The multicellular root template drawings show the auxin distribution at time point 5 min predicted by the mathematical model (I). The graphs show the predicted change in epidermal auxin levels in the upper versus the lower part of the root tip over time, calculated as the average auxin concentration in the transition-zone epidermal cells (J). (A, E, G) Three biological replicates were performed with similar results. Source data are available online for this figure.

evaluated the impact on the incorporation of HA:UBQ1 when these lysines (K) were replaced with arginines (R), by generating AUX1[3K>R] (mutating amino acids K261, K264, and K266) and AUX1[5K>R] (mutating amino acids K261, K264, K266, K339, and K347) protein variants. This revealed that the AUX1[3K>R] variant was less ubiquitinated than AUX1 (Fig. 4B,C; Appendix Fig. S18). Since the AUX1[5K>R] variant, compared to AUX1[3K>R], did not display an additional decrease in ubiquitination, we concluded that K261, K264, and K266 are the major target sites for ubiquitination, and we therefore focused on the AUX1[3K>R] variant for subsequent analyses. While SLOMO-mediated AUX1 ubiquitination was largely absent on the AUX1[3K>R] variant (Fig. 4D,E), the AUX1[3K>R] variant did not result in highly altered AUX1 levels (Fig. 4F,G; Appendix Fig. S19). Furthermore, the AUX1[3K>R] variant could only partially rescue the *aux1-22* mutant (Fig. 4H; Appendix Fig. S20). These results suggest that the AUX1[3K>R] variant is, at least partially, impaired in its activity. In addition, the trafficking and/or degradation of AUX1[3K>R]:YFP was not obviously affected compared to AUX1:YFP (Appendix Fig. S21), further suggesting that the K261, K264, and K266 are not impacting (vacuole-mediated) degradation.

We therefore speculated that the AUX1 ubiquitination on K261, K264, and K266 could affect AUX1 auxin transport properties. To explore this further, we used the AlphaFold2-predicted AUX1 structure to analyze the impact of the K > R mutations (Appendix Fig. S22). The root-mean-square deviation (RMSD) from the in silico molecular dynamics (MD) simulation showed that the motility of the AUX1[3K>R] variant is reduced (Appendix Fig. S22). The root-mean-square fluctuation (RMSF) points out that several AUX1 transmembrane helices are restricted in the AUX1[3K>R] variant, such as the helices near the mutated region including the 233–253 and 267–287 residues (Fig. 5A). These residues are crucial for the functionality of membrane channels and transporters (Lefoulon, 2021; Shaikh et al, 2013; Sato et al, 2023) that switch between open and closed states by having highly flexible transmembrane helices. This was further confirmed through auxin

transport assays using *N. benthamiana* protoplasts, where auxin influx activity was dramatically impaired in the AUX1[3K>R] variant compared with the AUX1 wild-type (Fig. 5B; Appendix Fig. S23), without affecting AUX1 protein levels and plasma membrane localization (Appendix Fig. S24). Furthermore, co-expression with *mCherry:SLOMO* in *N. benthamiana* did not visibly alter the plasma membrane localization levels of AUX1:YFP and AUX1[3K>R]:YFP compared to the control infiltrations (Appendix Fig. S24). In addition, while *pAUX1::AUX1:YFP* was able to partially restore sensitivity to 2,4-D, this was not the case for *pAUX1::AUX1[3K>R]:YFP* (Fig. 5C). Taken together, this suggests that replacing the lysine with an arginine at K261, K264, and K266, and thus preventing ubiquitination, impairs AUX1 transport activity.

Taken together, our findings reveal a mechanistic framework in which SLOMO modulates auxin transport by influencing both the abundance and activity of the auxin influx carrier AUX1, likely at distinct (or partially overlapping) lysine residues. We demonstrate that SLOMO controls auxin transport through affecting AUX1 protein levels, likely via ubiquitin-dependent processes, and through regulating AUX1 activity via non-proteolytic ubiquitination of specific lysine residues, namely K261, K264, and K266, likely via SLOMO. Nevertheless, both mechanisms might overlap in controlling rapid changes in AUX1 activity versus an overall change in AUX1 abundance.

It was previously suggested that PINs modulate the velocity of auxin transport, while AUX1 primarily defines spatial auxin accumulation across tissues (Band et al, 2014). A higher level of auxin will lead to a faster regulation of—at least—the transcriptional auxin response (Weijers and Wagner, 2016; Das et al, 2021). However, fine-tuning AUX1 transport capacity becomes a critical point of regulation during rapid processes, such as gravitropism. In support of earlier genetic evidence implicating SLOMO in auxin transport (Lohmann et al, 2010), we now show that SLOMO plays a dual regulatory role by adjusting AUX1 levels and its activity. Since an overall change in root tip AUX1 abundance alone does not fully account for the increased bending rates observed during gravitropic

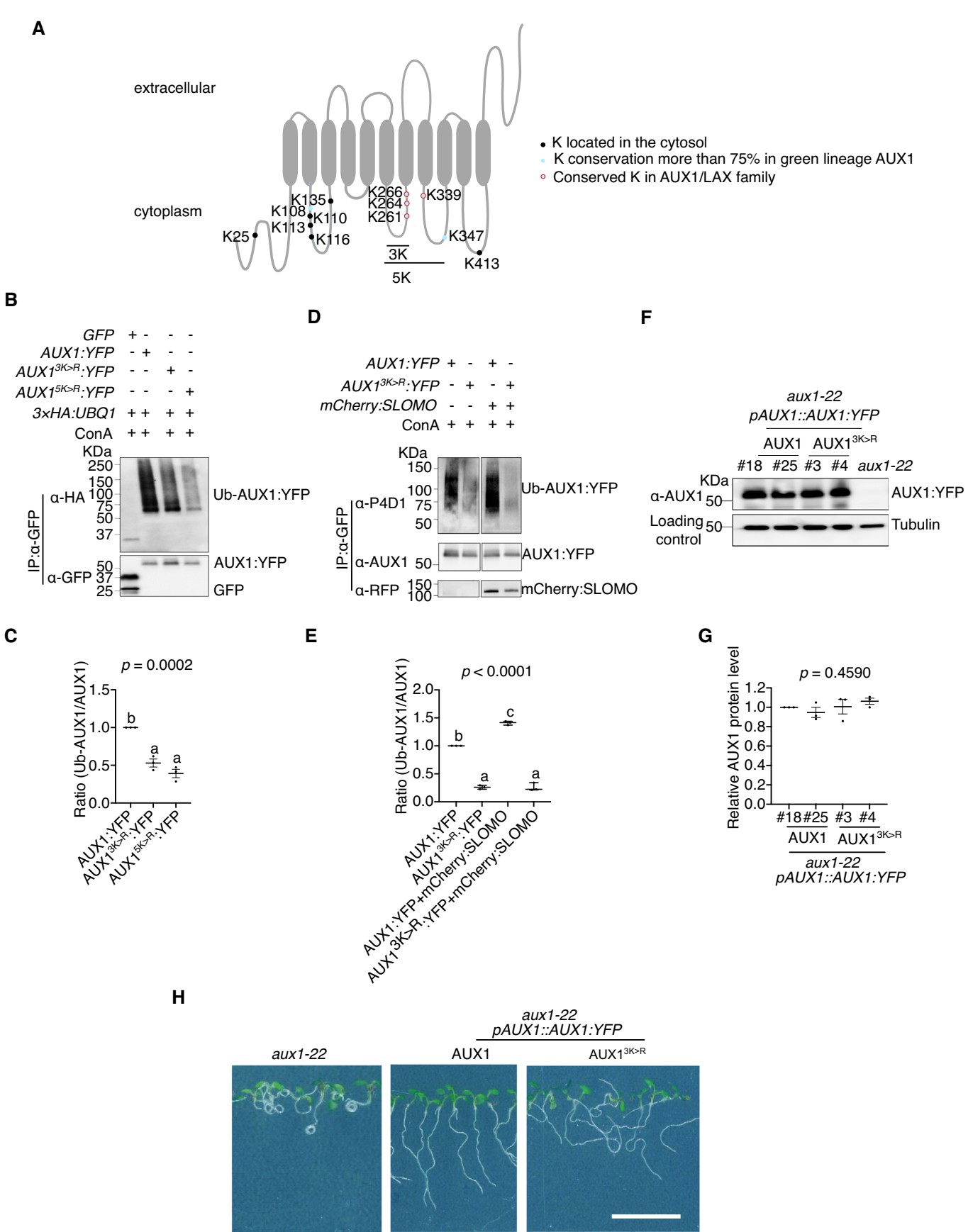

◄  **Figure 4.   K261, K264, and K266 are potential AUX1 ubiquitination sites.**

(A) Topology of AUX1. Lysines (K) are labeled as indicated. (B, C) Representative Western blot detecting Ub-AUX1 following HA:UBQ1 IP upon transient expression of *AUX1*, *AUX1³ᴷ>ᴿ* or *AUX1⁵ᴷ>ᴿ* with HA:UBQ1 in *N. benthamiana*. All the constructs are under the control of 35S promoter. Immunoprecipitation (IP) products precipitated by the α-HA antibody. IP products were probed with an anti-HA or anti-GFP antibody (B). The graph shows the quantification of the ub-AUX1/AUX1 ratio. ub-AUX1 was detected using α-HA, and AUX1 was detected using α-AUX1. The graph represents three independent biological replicates (individual dots) with the standard error of the mean. Different letters denote significant differences (*P* < 0.05) based on one-way ANOVA with Tukey's HSD. The *P* value among all variants (*P* = 0.0002) is shown at the top of the graph (C). (D, E) Representative Western blot detecting AUX1 ubiquitination following AUX1:YFP IP upon transient expression of *AUX1* or *AUX1³ᴷ>ᴿ* with SLOMO in *N. benthamiana*. IP products precipitated by the anti-GFP antibody. Ub-AUX1:YFP was detected with α-P4D1 antibody, AUX1-YFP was detected with α-AUX1 antibody, and mCherry:SLOMO was detected with α-RFP antibody (D). The graph shows the quantification of the ub-AUX1/AUX1 ratio. ub-AUX1 was detected using α-P4D1, and AUX1 was detected using α-AUX1. The graph represents three independent biological replicates (individual dots) with the standard error of the mean. Different letters denote significant differences (*P* < 0.05) based on one-way ANOVA with Tukey's HSD. The *P* value among all combinations (*P* < 0.0001) is shown at the top of the graph. (F, G) Representative western blot detecting AUX1 protein level in *aux1-22* lines expressing *AUX1* or *AUX1³ᴷ>ᴿ* following AUX1:YFP IP. AUX1-YFP was detected with α-AUX1 antibody (F). Graph of protein quantification (with tubulin as reference) shows three independent biological replicates (individual dots) with standard error of the mean (G). Different letters denote significant differences (*P* < 0.05) based on one-way ANOVA with Tukey's HSD. The *P* value among all genotypes (*P* = 0.4590) is shown at the top of the graph. (H) Representative image of *aux1-22* expressing *AUX1* or *AUX1³ᴷ>ᴿ*. Scale bar = 1 cm. (B, D, F) Three biological replicates were performed with similar results. Source data are available online for this figure.

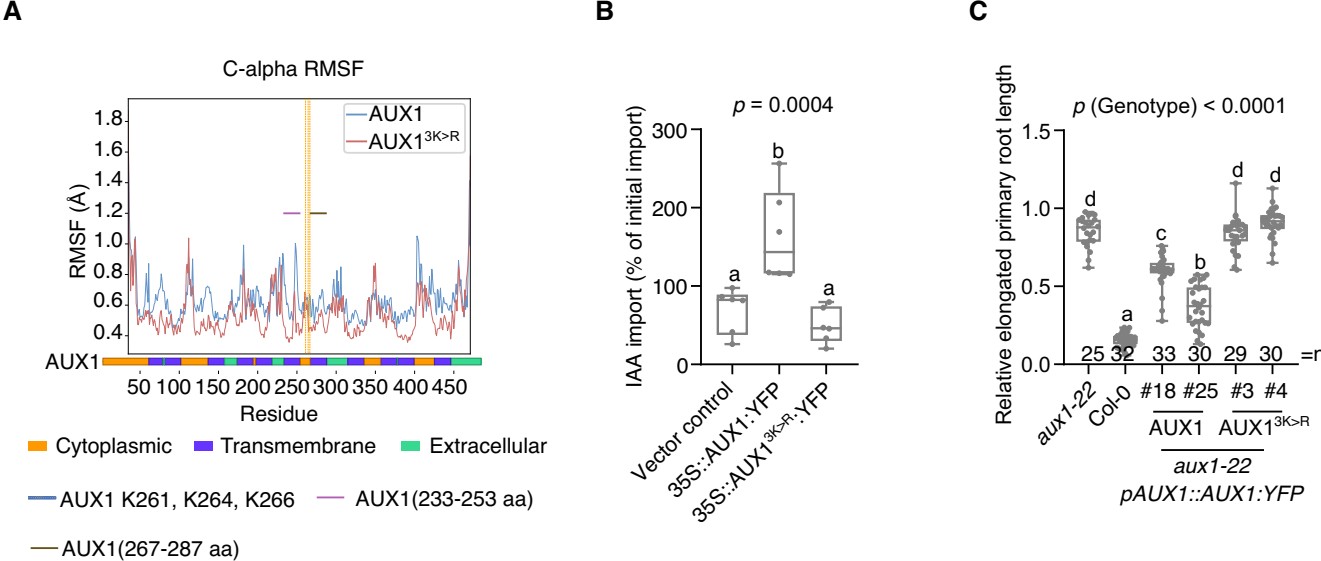

**Figure 5.   Key lysines determine AUX1 transport activity.**

(A) The root-mean-square fluctuation (RMSF) of the alpha carbon (Cα) of AUX1 residues within a 50-ns molecular dynamics simulation. The color codes below the diagram indicate the AUX1 topology. (B) Quantification of indole-3-acetic acid (IAA) import in protoplasts prepared from *N. benthamiana* leaves transfected with *35S::AUX1:YFP* or *35S::AUX1:YFP³ᴷ>ᴿ*. Box plots with individual experiment data points represent the distribution of IAA import for six biological replicates. (C) Relative newly grown primary root upon transfer to 100 nM 2,4-D for *aux1-22* expressing *AUX1* or *AUX1³ᴷ>ᴿ* (2,4-D treatment/DMSO). The number of individually measured seedlings (*n*) is indicated above the X axis. For the graphs (B, C), the box spans from the 25th percentile to the 75th percentile of the data. A horizontal line inside the box marks the median (50th percentile). Whiskers extend from the minimum to the maximum. The letters indicate significant differences based on one-way ANOVA with Tukey's HSD (*P* < 0.05). The *P* value among the genotypes (*P* < 0.0001) or constructs (*P* = 0.0004) is shown at the top. Source data are available online for this figure.

responses, additional layers of regulation are required. Therefore, we propose a regulation on AUX1 activity, which also seems to be regulated through ubiquitination and in which SLOMO is likely involved. This is supported by mutagenesis of the identified lysine residues to arginine, thereby preventing ubiquitination, which results in reduced AUX1 transport activity without affecting its stability. Furthermore, the role of SLOMO in this process is supported by strongly reduced ubiquitination in the *slomo* mutant, and the largely absent SLOMO-mediated AUX1 ubiquitination on the AUX1³ᴷ>ᴿ variant. Although we cannot fully rule out an effect on AUX1 structure, this points to a functional role for

ubiquitination in enhancing AUX1 activity, rather than targeting it for degradation, revealing a novel, non-proteolytic role of SLOMO-mediated ubiquitination of AUX1. While ubiquitination may not be strictly required for AUX1 function, it appears that replacing lysines with arginines decreases AUX1 transport activity, and ubiquitination could contribute to full AUX1 transport activity. This is also in line with the *35S::SLOMO* and/or *slomo* loss-of-function lines not fully capturing the strong *aux1* loss-of-function phenotypes. It is possible that the dramatic effect on AUX1 transport activity in the protoplast system is less penetrant in a whole-seedling context, where other factors also contribute to and

overcome the loss of specific auxin transport routes. While we identified putative AUX1 ubiquitination sites that impact AUX1 activity and not AUX1 levels when mutated, SLOMO must affect AUX1 levels either at other so far unknown ubiquitination sites and/or indirectly via regulating another F-box protein. At the moment, we do not know which lysine residues on AUX1 are targeted for this and which other F-box protein could be involved.

Our spatial analyses further revealed that SLOMO activity in the root cap and epidermis is sufficient to confer AUX1 regulation during gravitropic response, suggesting that only a subset of AUX1-dependent auxin transport is under SLOMO control and that AUX1 pools in internal tissues, such as the stele, are unaffected by this pathway. This spatial restriction may contribute to differences in auxin flux asymmetry during gravitropic responses, and this might also explain why the *slomo* mutant displays a partial gravitropic defect that is milder than the complete agravitropic phenotype of *aux1* mutants. Nonetheless, *slomo* exhibits a clear defect in gravitropic bending, indicating a significant role for SLOMO in this process.

To fully incorporate a role for SLOMO in ubiquitin-mediated AUX1 activity on the lower side of the root tip, another regulatory layer is required at the level of SLOMO, and a clear separation between the effect on AUX1 activity versus AUX1 abundance. At the moment, we can only speculate on the exact SLOMO-controlled mechanism that establishes differential activity of AUX1 on the lower versus the upper side of the gravity-stimulated root tip. The small asymmetry observed in SLOMO levels between the upper and lower sides of the root post-gravistimulation might underlie this asymmetry. While this could strengthen the conclusion that SLOMO can asymmetrically impact AUX1 activity (increased SLOMO-mediated AUX1 ubiquitination to promote AUX1 influx activity), we do not want—given the very small difference—put too much emphasis on this. In addition, since SLOMO also affects AUX1 abundance, this might also lead to a reduction in AUX1 abundance on the lower root side, and reduced auxin transport. It could imply that at this level, the control over AUX1 abundance by SLOMO is less critical than the control over activity, or both roles of SLOMO are differently regulated, requiring additional components. It is possible that SLOMO-mediated regulation of AUX1 activity occurs on a faster timescale than changes in protein abundance, since at 6 h after gravistimulation, the asymmetric distribution of GFP:SLOMO is less pronounced. In addition, additional regulatory components - such as spatially distinct SCF complex assemblies or activity or deubiquitinating enzymes (DUBs)—might mediate the fine-tuning of AUX1 ubiquitination. SLOMO has been shown to interact with the DUBs UBP12 and UBP13 (Zhu et al, 2024). However, *ubp12-2* mutants, in which the levels of both *UBP12* and *UBP13* transcripts are downregulated (Cui et al, 2013; Vanhaeren et al, 2020), did not exhibit striking defects in root gravitropism (Appendix Fig. S25), suggesting that other DUBs or regulatory mechanisms may be involved.

In conclusion, our biochemical analyses, auxin transport measurements, genetic analyses, and computational modeling converge on a model in which SLOMO-mediated ubiquitination fine-tunes AUX1 function during gravitropic signaling. Notwithstanding direct evidence that SLOMO-mediated ubiquitination of K261, K264, and K266 affects AUX1 activity is lacking at the moment, we propose that SLOMO promotes AUX1 activity via ubiquitination at K261, K264, and K266, and that this process is spatially and temporally modulated in response to gravistimulation. This modulation may be counterbalanced by an, as yet unidentified, DUB or differential SCF complex activity at the different sides of the root tip, ultimately contributing to precise auxin gradient formation and directional root growth.

## Methods

**Reagents and tools table**

| Reagent/resource | Reference or source | Identifier or catalog number |
|---|---|---|
| **Experimental models** | | |
| Arabidopsis: *slomo-3, slomo-4, slomo-5, slomo-3 pSLOMO::GFP:SLOMO, slomo-4 pSLOMO::GFP:SLOMO, slomo-5 pSLOMO::GFP:SLOMO* | Lohmann et al, 2010; Zhu et al, 2024 | N/A |
| Arabidopsis: *pDR5::GUS* | Ulmasov et al, 1997 | N/A |
| Arabidopsis: *aux1-22* and *aux1-22 pAUX1::AUX1:YFP* | Swarup et al, 2004 | N/A |
| Arabidopsis: Col-0 *35S::mCherry:SLOMO* | This study | N/A |
| Arabidopsis: *aux1-22 pAUX1:: AUX1³ᴷ>ᴿ:YFP* | This study | N/A |
| Arabidopsis: *ProLAX1:LAX1-VENUS* and *ProLAX2:LAX2-VENUS* | Péret et al, 2012 | N/A |
| Arabidopsis: *pSKS15::GFP:SLOMO in slomo-3, pGLL23:: GFP:SLOMO in slomo-3, pAT1G14220:: GFP:SLOMO in slomo-3* | In this study | N/A |
| Arabidopsis: *upb12-2* | Vanhaeren et al, 2020 | N/A |
| *Escherichia coli*: DH5α | Thermo Fisher Scientific | Cat # 18265017 |
| *Agrobacterium tumefaciens* strain: C58C1 | Clough and Bent, 1998 | N/A |
| Yeast: THY.AP4 | Obrdlik et al, 2004 | N/A |
| **Recombinant DNA** | | |
| *35S::mCherry:SLOMO, 35S::mCherry:SLOMO-Decoy* | Zhu et al, 2024 | N/A |
| *35S::MYRI-TagBFP2FRB* | Winkler et al, 2021 | N/A |
| *pAUX1::AUX1:YFP* | Swarup et al, 2004 | N/A |
| *35::AUX1:YFP* | This study | N/A |
| *pAUX1:: AUX1³ᴷ>ᴿ:YFP* | This study | N/A |
| *35S:: AUX1³ᴷ>ᴿ:YFP* | This study | N/A |
| *35S:: AUX1⁵ᴷ>ᴿ:YFP* | This study | N/A |
| *35S:LAX1:VENUS* and *35S:LAX2:VENUS* | This study | N/A |
| *pMetYC-Dest:AUX1* | This study | N/A |
| *pNX32-Dest:SLOMO* | This study | N/A |
| *pDonrP4P1r:PSKS15, pDonrP4P1r:GLL23, pDonrP4P1r:AT1G14220* | Wendrich et al, 2020 | N/A |

| Reagent/resource | Reference or source | Identifier or catalog number |
|---|---|---|
| *pSKS15::GFP:SLOMO, pGLL23:: GFP:SLOMO, pAT1G14220:: GFP:SLOMO* | This study | N/A |
| *35S::TagRFP:SLOMO* | This study | N/A |
| **Antibodies** | | |
| α-GFP | Miltenyi Biotech | Cat # 130-131-646 |
| α-HA | abcam | Cat# ab1190 |
| α-AUX1 | Agrisera | Cat# AS16 3957 |
| α-RFP | ChromoTek | Cat# 6g6 |
| α-P4D1 | Santa Cruz | Cat# sc-8017 |
| α-Mouse | Sigma-Aldrich | Cat# A9044 |
| Anti-Goat | Agilent Dako | Cat# P0449 |
| α-Tubulin | Sigma-Aldrich | Cat# T5168 |
| GFP-Trap®_MA | Chromotek | Cat# gtma-500 |
| HA-Trap®_MA beads | Thermo Fisher Scientific | Cat# 88837 |
| **Oligonucleotides and other sequence-based reagents** | | |
| PCR and qPCR primers | In this study | N/A |
| **Chemicals, enzymes, and other reagents** | | |
| NPA (N-1-naphthylphthalamic acid) | Sigma-Aldrich | Cat# 86-87-3 |
| NOA (1-Naphthoxyacetic acid) | Sigma-Aldrich | Cat# 120-23-0 |
| 2,4-D (2,4-Dichlorophenoxyacetic acid) | Sigma-Aldrich | Cat# 94-75-7 |
| ConA | Santa Cruz | Cat# sc-203007A |
| MG132 | Sigma-Aldrich | Cat# 133407-82-6 |
| Bortezomib | Sigma-Aldrich | Cat# 179324-69-7 |
| X-Gluc | Thermo Fisher Scientific | Cat# R0852 |
| FM4-64 | Thermo Fisher Scientific | Cat# T13320 |
| propidium iodide (PI) | Sigma-Aldrich | 25535-16-4 |
| 3H-IAA and 14C-BA | Henrichs et al, 2012 | N/A |
| **Software** | | |
| Fiji ImageJ | Schindelin et al, 2012 | N/A |
| Image Lab | Bio-Rad | N/A |
| CLC Main Workbench 8 | CLC Bio-Qiagen, Aarhus, Denmark | N/A |
| NAMD 3.0b6 | Phillips et al, 2005 | N/A |
| Chimera X | Meng et al, 2023 | N/A |
| GraphPad Prism 8.0.1 | www.graphpad.com | N/A |

| Reagent/resource | Reference or source | Identifier or catalog number |
|---|---|---|
| **Other** | | |
| Dicots PLAZA 5.0 | Van Bel et al, 2022 | N/A |
| CHARMM36m | Best et al, 2012 | N/A |
| CHARMM-GUI | Best et al, 2012; Jo et al, 2008 | N/A |
| AlphaFold | Varadi et al, 2022 | N/A |
| MDAnalysis | Michaud-Agrawal et al, 2011 | N/A |

## Plant materials and growth conditions

All *Arabidopsis thaliana* plants used in this study were in the Col-0 reference accession genetic background and referred to as wild-type. The following *Arabidopsis thaliana* lines were used: *slomo-3, slomo-4, slomo-5, slomo-3 pSLOMO::GFP:SLOMO, slomo-4 pSLO-MO::GFP:SLOMO, slomo-5 pSLOMO::GFP:SLOMO* (Lohmann et al, 2010; Zhu et al, 2024); Col-0 *35S::mCherry:SLOMO, aux1-22 pAUX1::AUX1:YFP 35S::mCherry:SLOMO; pDR5::GUS* (Ulmasov et al, 1997); *aux1-22* and *aux1-22 pAUX1::AUX1:YFP* (Swarup et al, 2004); *ubp12-2* (Vanhaeren et al, 2020). For the phenotypic assay, Arabidopsis seeds were sown on ½ Murashige and Skoog (MS) growth medium containing 1% sucrose (per litre: 2.15 g of MS salts, 0.1 g of myo-inositol, 0.5 g of MES, 10 g of sucrose, and 8 g of plant tissue culture agar; pH 5.7). For the NPA, NOA, and 2,4-D treatments to Col-0 and *slomo-3*, NPA and NOA were dissolved in DMSO. The ½ MS medium was then supplemented with 5 µM NPA or 1 µM NOA, 50 nM 2,4-D, or an equal volume of DMSO as Mock. For AUX1 protein level and ubiquitination assays with *Arabidopsis thaliana* plants, seedlings were grown on sugar-free ½ MS plates. For ConA, MG132, and Bortezomib (BTZ) treatment, seedlings grew on ½ MS sugar-free medium under continuous light conditions for 5 days after germination, then the seedlings were transferred to liquid ½ MS sugar-free medium with DMSO, and 1 µM ConA, 50 µM MG132, and 40 µM BTZ, respectively, treated for 5 h in darkness.

## Gravitropism assay

For the gravitropism treatment, seedlings were vertically grown on ½ MS medium under continuous light conditions for 3 days after germination (DAG), and subsequently, the seedlings were transferred, with their root tips positioned vertically, to a new ½ MS plate and then rotated 90 degrees, and were kept in the dark during the gravitropic experiment (time course of 6 h gravistimulation). Images were taken with a Canon scanner and root tip orientation measurements were performed using Fiji ImageJ (https://imagej.net/Fiji). Root angles are indicated as the deviation from 0°. The angles of root curvature were grouped in 18 sectors of 20° (or twelve sectors of 30°) around the circle, and the length of each bar represents the percentage of seedlings showing direction of root growth within that sector.

## Live tracking of moving samples in confocal microscopy for vertically grown roots

The tracking was adapted from a published method with some modifications (Vukašinović et al, 2025). Seedlings were grown vertically on MS growth medium at 21 °C under continuous light for 5 days. The seedlings were transferred onto agar slabs containing MS growth medium in chamber slides (Ibidi μ-Slide 1 Well) sealed with surgical tape. The seedlings were then grown at 21 °C under continuous light for an additional 1 h before imaging. The live tracking of root tips was performed using a vertical ZEISS LSM900 microscope with the objective lens Plan-Apochromat M27 20×/0.8 n.a. Seedlings per condition were turned 90° to apply a gravity stimulus before tracking up to 6 h using TipTracker (von Wangenheim et al, 2017). In total, 15 to 20 Z-stacks with 2 μm interval were captured. A 488 nm excitation laser and an emission detection bandwidth of 493–540 nm were used. The images were subjected to Z maxima projection before intensity measurements using Fiji ImageJ (https://imagej.net/Fiji).

## Phenotyping of primary and lateral roots and meristem length

For primary and lateral root phenotyping, Col-0 and *slomo-3* seedlings were grown at 21 °C under continuous light for 12 days. Subsequently, the plates were scanned, the primary root length was measured using ImageJ, and the number of emerged lateral roots was counted. For root meristem length analysis, seedlings were grown under continuous light for 7 days, stained with propidium iodide (PI), and imaged using a Leica SP8 confocal microscope, and the meristem zone length was measured by ImageJ.

## 2,4-D treatment for *aux1-22* complementing lines

Seedlings were grown vertically on MS growth medium (per litre: 4.3 g MS salts, 1% sucrose, 1% plant tissue culture agar, pH to 6.0 with KOH) for 3 days after germination before being transferred to a new MS containing 100 nM 2,4-D for additional 3 days. The elongated primary root was measured by using Fiji ImageJ.

## GUS staining assay

Three days-after-germination-old *pDR5::GUS* and *pDR5::GUS slomo-3* seedlings after the gravistimulation were fixed for 30 min in ice-cold 90% (v/v) acetone and rinsed with NT buffer (100 mM Tris-HCl (pH = 7.0), 50 mM NaCl), incubated in X-Gluc (500 μg ml$^{-1}$ 5-bromo-4-chloro-3-indolyl-β-D-glucuronide) solution (500 μl 100 mM K$_3$[Fe(CN)$_6$] + 600 μl X-Gluc + 28.8 ml NT-buffer), and incubated at 37 °C in the dark for 3 h. The reaction was stopped by rinsing with NT-buffer, and then the seedlings were mounted in 80% lactic acid on a slide and imaged with an Olympus microscope.

## Confocal microscopy

Seedlings were mounted in Milli-Q water between the slide and cover glass for imaging using a Zeiss inverted LSM710 confocal laser scanning microscope, equipped with a LD-Plan Neofluar ×40/ 0.6 Korr M27 or C-Apochromat ×40/1.20 W Korr M27. A 488 nm laser excitation (at 2% power) of a 20 mW argon laser (LASOS, Jena, Germany) and a spectral detection bandwidth of 493–532 or 493–579 nm were used for detecting eGFP. For propidium iodide (PI) staining, whole seedlings were mounted in PI solution. The PI signal was detected using a 561 nm laser excitation (at 2% power), together with a spectral detection bandwidth of 596–645 nm. Fluorescence intensity of AUX1:YFP signal was measured by Fiji ImageJ (https://imagej.net/Fiji).

## Microsomal fraction preparation

Isolation of microsomal fractions was performed as previously described (Abas and Luschnig, 2010) with minor modifications. Seedlings were ground in liquid nitrogen and resuspended in ice-cold sucrose buffer [100 mM Tris-HCl, pH 7.5, 810 mM sucrose, 5% (vol/vol) glycerol, 10 mM EDTA, pH 8.0, 10 mM EGTA, pH 8.0, 5 mM KCl, and complete protease inhibitor mixture and the PhosSTOP phosphatase inhibitor mixture (both from Roche)]. The homogenate was transferred to a pre-cold 5% (vol/vol) polyvinyl-polypyrrolidone pellet, mixed, and then kept on ice for 5 min. Samples were then centrifuged for 5 min at 600× g at 4 °C, and the supernatant was collected. This extraction step was repeated two more times. The clear supernatant was combined with the same volume of distilled water and centrifuged at 4 °C for 2 h at 21,000× g to pellet microsomes. The pellets were washed with 1 ml of wash buffer (10 mM Tris-HCl, pH 7.5) and centrifuged at 4 °C for 30 min at 21,000× g and stored at −70 °C until further use.

## Co-immunoprecipitation in *Arabidopsis thaliana*

For co-immunoprecipitation (co-IP) and ubiquitination assays, re-suspension of the membrane pellets with extraction buffer [50 mM Tris-HCl, pH = 7.5, 150 mM NaCl, 1 mM EDTA, 1 mM EGTA, 10% SDS, and protease inhibitor (Roche), to which 1 mM of PMSF and 10 mM of iodoacetamide were freshly added]. This suspension was centrifuged at 4 °C for 30 min at 21,000×g to obtain a clear supernatant. The solubilized microsomal proteins were first diluted 10x with extraction buffer containing without SDS and then incubated with 25 μL pre-equilibrated GFP-Trap®_MA beads (Chromotek) and rotated for 2 h at 4 °C to maximize protein binding. The samples were cleared on a magnetic separation rack and remove supernatant. To remove unspecific binding, the beads were washed in 3 × 1 mL wash buffer [25 mM Tris-HCl, PH = 7.5, 250 mM NaCl]. Finally, 1× sample buffer (Bio-Rad) was added to elute protein from the beads.

## Western blot

For SLOMO protein abundance analysis, protein samples were heated at 70 °C for 10 min. For AUX1 protein detection, protein samples were heated at 28 °C for 45 min, then separated on 4–20% SDS-PAGE stain-free protein gel (Bio-Rad Laboratories, Inc., USA), followed by transferring onto a Trans-Blot® Turbo™ Mini PVDF Transfer Packs (Bio-Rad Laboratories, Inc., USA). For blocking and antibody dilutions, 5% milk powder in TBST solution was used. For protein detection, the following antibodies were used: monoclonal α-GFP horseradish peroxidase coupled (1:5000; Miltenyi Biotech), α-HA (1:5000; abcam), α-AUX1 (1:5000, Agrisera), α-RFP (1:2000, ChromoTek), α-P4D1 (1:2000; Santa

Cruz), mouse IgG HRP-linked whole antibody (1 : 10,000, Sigma-Aldrich, USA), Rabbit Anti-Goat Immunoglobulins/HRP (1:10,000, Agilent Dako), and α-Tubulin (1:5000, Sigma-Aldrich). The proteins were detected by ChemiDoc™ MP Imaging System (Bio-Rad Laboratories, Inc., USA). When required, the membranes were stripped using stripping buffer (1:1–1.5 ml 10% SDS: 1.5 ml 100 mM Glycine mix) for 90 s, were washed 3×5 min with T-BST, and then again subjected to the procedure described above.

## Quantification of western blots

Quantification for relative protein intensity was measured by Image Lab (Bio-Rad). A single band was designated as the reference volume by selecting the Reference Volume checkbox in the Volume Properties dialog box. In the Volume Table tab of the analysis table, the Relative Quantity column displays relative quantities. These values represent the ratio of the background-adjusted volume to the background-adjusted reference volume. All other sane volume bands are expressed as numerical values relative to the reference volume. The value of the reference band is 1.00. A value greater than 1.00 indicates that the volume exceeds the reference intensity, while a value less than 1.00 indicates it is smaller than the reference intensity.

## Ubiquitination assay in Arabidopsis thaliana

Col-0 *and slomo-3* background of *pAUX1::AUX1:YFP aux1-22* seedlings were grown at 21 °C under continuous light for 11 days, after which the seedlings were harvested for microsomal fraction preparation and protein microsomal protein extraction. Same amount of total protein used for immunoprecipitation with GFP-Trap®_MA beads (Chromotek). AUX1 protein samples were incubated at 28 °C for 45 min. The same amount of AUX1:YFP of input and IP samples loaded for western blotting was adjusted accordingly. Anti-AUX1 antibody was used to detect both AUX1 and ubiquitinated AUX1 (ub-AUX1). Anti-P4D1 antibody was used to detect ub-AUX1.

## *Nicotiana benthamiana* infiltration

For transient co-agroinfiltration of *Nicotiana benthamiana* leaves, *Agrobacteria* containing the indicated constructs and P19 were grown in 5 ml LB supplemented with appropriate antibiotics at 28 °C for 1–2 days. Then 500 µl was inoculated in 10 ml LB supplemented with 10 mM MES pH 5.6, 10 µM acetosyringone as well as the antibiotics and incubated at 28 °C overnight. The pellet was spun down and resuspended with infiltration buffer (10 mM MgCl₂, 10 mM MES [pH 5.6], 100 µM acetosyringone) to a final OD₆₀₀ of 1.0. Equal volumes of three agrobacteria with selected constructs were mixed well and the mix was infiltrated into 5–6 weeks old tobacco leaves using a syringe. The signal was checked by confocal microscopy after 72 h to make sure the protein was expressed well. For the ConA treatment, 1 µM ConA in infiltration buffer was infiltrated into the same *Nicotiana benthamiana* leaves 16 h before harvesting. For the observation of AUX1:YFP, AUX13K > R:YFP and mCherry:SLOMO localization in *N. benthamiana* leaves, *Agrobacteria* containing the indicated constructs and P19 were grown on YEB plates supplemented with appropriate antibiotics at 28 °C for 2 days, respectively. Then we took a swath of *Agrobacteria* from plates and resuspended with infiltration buffer (10 mM MgCl₂, 10 mM MES [pH 5.6], 100 µM acetosyringone)

to a final OD600 of 1. Equal volumes of three *Agrobacteria* with selected constructs were mixed well and the mix was infiltrated in 5–6 weeks old *N. benthamiana* leaves using a syringe.

## Co-immunoprecipitation (co-IP) and ubiquitination assay in *Nicotiana benthamiana*

Total proteins were extracted with buffer containing 150 mM Tris·HCl, pH 7.5, 150 mM NaCl, 10% glycerol, 100 mM EDTA, 1 mM sodium molybdate, 1 mM NaF, 10 mM DTT, 1% NP-40, 1 mM PMSF, and EDTA-free protease inhibitor mixture complete (Roche). The protein concentration was measured using the Qubit™ Protein Assay Kit (Qubit™ Protein Assay Kit), and an equal amount of total protein was used for immunoprecipitation. 25 µl of pre-equilibrated GFP-Trap® MA beads (ChromoTek) or HA-Trap®_MA beads (Thermo Fisher) was prewashed 3 times with 700 µl wash buffer (20 mM Tris·HCl, pH 7.5, 150 mM NaCl and 0.5% NP-40). The protein homogenates were incubated with GFP-Trap® MA beads or HA-Trap®_MA beads and rotated for 2 h at 4 °C to maximize protein binding. Subsequently, separating protein homogenates and beads utilizing a magnet, the beads were washed three times with wash buffer (20 mM Tris-HCl pH 7.5, 150 mM NaCl, and 0.5% NP-40). Finally, 1× sample buffer (Bio-Rad) was added to elute protein from the beads.

## qPCR analyses

Three biological replicates were performed for each condition. Total RNA was extracted with the RNeasy Mini Kit (Promega) according to the manufacturer's instructions. DNA digestion was done on columns with RNase-free DNase I (Promega). The iScript cDNA Synthesis Kit (Biorad) was used for cDNA synthesis from 1 µg of RNA. qRT-PCR was performed on a LightCycler 480 (Roche Diagnostics) in 384-well plates with LightCycler 480 SYBR Green I Master (Roche) according to the manufacturer's instructions. Two housekeeping genes, ACTIN and the EF1α were used for normalization of the expression level of the tested genes. All the primers are listed in Appendix Table S1.

## Generation of constructs and transgenic lines

*35S::mCherry:SLOMO, 35S::mCherry:SLOMO*-Decoy, *SLOMO* promotor, *AUX1* promotor and 35S::MYRI-TagBFP2FRB were described previously (Zhu et al, 2024; Péret et al, 2012; Winkler et al, 2021). The DNA sequences of *AUX1:YFP*(116 aa) and *AUX1³ᴷ>ᴿ:YFP*(116 aa), referred to as AUX1:YFP and AUX1³ᴷ>ᴿ:YFP, respectively, were synthesized by Twist Bioscience, and the synthesized sequences were assembled into the entry vector pEN-L1-AG-L2 and then ligated to the relevant destination vector via a gateway cloning system. For the *35S:LAX1:VENUS* and *35S:LAX2:VENUS* constructs, *LAX1:VENUS* and *LAX2:VENUS* fragments were amplified from seedlings of the *ProLAX1:LAX1-VENUS* and *ProLAX2:LAX2-VENUS* lines (Péret et al, 2012), respectively. These fragments were first cloned into the pDONR221 entry vector, and subsequently recombined with pEN-L4-2-L1 into the destination vector pB7m24GW,3. Plant vectors were transformed in *Agrobacterium tumefaciens* C58C1 using a freeze–thaw method (Clough and Bent, 1998). pAUX1::AUX1:YFP and pAUX1::AUX1³ᴷ>ᴿ:YFP transformed to *aux1-22*; *35S::mCherry:-SLOMO* transformed to Col-0 and *aux1-22 pAUX1::AUX1:YFP*

respectively performed using floral dip method (Clough and Bent, 1998). To generate the constructs for the split-ubiquitin yeast assay, the CDS sequences of *AUX1* and *SLOMO* were amplified using Col-0 cDNA as a template. *AUX1* without a stop codon was then assembled into the entry vector pDONR221 and subsequently into the destination vector pMetYC-Dest, while SLOMO was assembled into pEN-L1-AG-L2 and then into the destination vector pNX32-Dest. pNubWtXgate was used as a positive control (Obrdlik et al, 2004). For the tissue-specific expression of SLOMO, based on the pFAST seed selection system, lateral root cap–specific (pDonrP4P1r:PSKS15) (Wendrich et al, 2020) and epidermis–specific (pDonrP4P1r:GLL23, pDonrP4P1r:AT1G14220) (Wendrich et al, 2020) promoter entry clones were assembled with pEN-L1-F-L1 and pDONR-P2rP3:SLOMO into the destination vector pB8m34GW-FAST using Gateway cloning. The resulting constructs were introduced into *Agrobacterium tumefaciens* strain C58 and transformed into *slomo-3* mutant via the floral dip method. T1 seeds were screened for GFP fluorescence in the seed epidermis under a fluorescence stereomicroscope.

## AUX1 plasma membrane localization assay

Seedlings were grown on ½ MS sugar-free medium under continuous light conditions for 5 days after germination. They were then transferred to liquid ½ MS sugar-free medium and treated with either DMSO or 1 μM ConA for 16 h in darkness. For visualization of AUX1:YFP, root tips were imaged using z-stack acquisition followed by maximum intensity projection. Relative fluorescence intensity (plasma membrane / cytosol) in individual cells was quantified using ImageJ software.

## Mathematical modeling

Simulations were based on the model of auxin dynamics presented previously (Mellor et al, 2020), which provides full details of the model assumptions, equations, and parameter values used. In brief, the model uses a multicellular root tip geometry extracted from a confocal image of a root tip, and distributions of PIN, AUX1/LAX, and plasmodesmata based on experimental observations. The modeling framework integrates these data into a system of linear ordinary differential equations (ODEs) for the auxin concentration within each cell and cell-wall compartment, which contain terms representing passive diffusion of protonated auxin across cell membranes, carrier-mediated transport of anionic auxin across cell membranes, auxin diffusion within the cell wall, auxin diffusion through plasmodesmata, auxin synthesis, and degradation. These ODEs are simulated using the Python ODE solver, and the matplotlib library is used to present the model solutions. To simulate gravitropic reorientation, we assume that at time $t = 0$, PIN3 and PIN4 in the columella cells are repositioned to the membrane on the lower side of the cells. Further simulations considered a change in AUX1 activity after reorientation and were produced by prescribing a different value for AUX1 activity on the upper/lower side of the root at time $t = 0$.

## AUX1 lysine sites conservation assay

The AUX1 (AT2G38120) protein sequence was downloaded from TAIR (https://www.arabidopsis.org/). The 560 AUX1 orthologous genes from 96 species were collected from Dicots PLAZA 5.0 by blasting (Van Bel et al, 2022). The AUX1 alignment was performed with CLC Main Workbench 8 (CLC Bio-Qiagen, Aarhus, Denmark).

## Molecular dynamics (MD) simulations and analysis

All-atom MD simulations were performed using NAMD 3.0b6 (Phillips et al, 2005) with the CHARMM36m force field (Best et al, 2012). Protein system building was performed using CHARMM-GUI (Best et al, 2012; Jo et al, 2008). The predicted AUX1 structure was downloaded from the AlphaFold protein structure database (Varadi et al, 2022) and trimmed from residue 35 to 471 to remove low-confidence protein tails. The wild-type or mutated AUX1 was inserted into a phosphatidylcholine (POPC) bilayer membrane and solvated with TIP3P water molecules (Jorgensen et al, 1983) at both sites of the bilayer lipid membrane in periodic rectangular boxes to a depth of at least 80 Å. The systems were minimized for 1000 steps and then simulated under the NVT (constant number of atoms, volume, and temperature) system at 295 K using a Langevin thermostat. Four independent 50 ns simulations of both wild-type and mutated AUX1 systems were performed with structure saving per 100 ps. The resulting simulation was analyzed using the Python package MDAnalysis (Michaud-Agrawal et al, 2011). The RMSD of C-alpha was calculated from the structure after minimization. The RMSF of C-alpha was calculated over all residues. The structural representations of the proteins were generated using Chimera X (Meng et al, 2023).

## Arabidopsis protoplast transformation

As previously described (Yoo et al, 2007), the detached leaves from 4-week-old Col-0 plants were peeled with label and scotch tapes followed by incubation with enzyme solution (0.4 M mannitol, 20 mM KCl, 20 mM MES-KOH [pH 5.7], 1% cellulase R-10, 0.25% macerozyme R-10, 10 mM CaCl2, 5 mM β-mercaptoethanol and 0.1% BSA) for 1 h under darkness. The protoplasts were washed 3 times with W5 solution (154 mM NaCl, 125 mM CaCl2, 5 mM KCl, 2 mM MES-KOH [pH 5.7] and 5 mM glucose) and then incubated in MMG solution (0.4 M mannitol, 15 mM MgCl₂ and 4 mM MES-KOH [pH 5.7]) on ice for 30 min. In total, $2 \times 10^4$ protoplasts in MMG solution were mixed with 10 μg *35S::AUX1:YFP*, *35S::TagRFP:SLOMO* and/or *35S::AUX1^{3K>R}:YFP* before PEG solution (40% PEG-4000, 0.2 mM mannitol and 0.1 mM CaCl2) treatment. The treatment was stopped by adding W5 solution. The PEG-treated protoplasts were washed three times with W5 solution and incubated under continuous light at 21 °C for 16 h before imaging.

## Auxin transport measurements

Simultaneous 3H-IAA and 14C-BA import into *Nicotiana benthamiana* and *Arabidopsis thaliana* mesophyll protoplasts was analyzed as described previously (Henrichs et al, 2012). *Nicotiana benthamiana* and *Arabidopsis* mesophyll protoplasts were prepared 4 days after *Agrobacterium*-mediated transfection of *35S::AUX1:-YFP* or 35S::AUX1^{3K>R}:YFP or 4 weeks after germination, respectively. Relative import into protoplasts is calculated from imported radioactivity into the protoplast as follows: (radioactivity in the

supernatant at time $t = 15$ min.) - (radioactivity in the protoplast at time $t = 0$) * (100%)/ (radioactivity in the protoplast at $t = 0$ min).

## Split-ubiquitin system in yeast

The haploid yeast strain THY.AP4 was co-transformed by heat shock with the Cub and Nub constructs of interest. For each co-transformation, 5 µl denatured Salmon Sperm carrier DNA (Invitrogen) and 40 µl yeast-competent cells were mixed. This was added to a tube containing 500 ng of each plasmid and incubated at 30 °C for 15 min. Next, 150 µl PLI solution (for 15 ml PLI solution: 12 ml 50% PEG 3350 solution, 1.5 ml 1 M lithium acetate solution, and 1.5 ml distilled water) was added and mixed thoroughly by pipetting. The mixture was then incubated at 30 °C for 30 min followed by an incubation at 42 °C for 15 min. The PLI solution was removed after 5 min of centrifugation at $3000 \times g$. The cells were resuspended in YPD and shaken for 2 h at 200 rpm at 30 °C. The yeasts were then plated out on SD/-Leu/-Trp plates and grown for 3 days at 30 °C. For each combination, 16 individual colonies were selected and grown in liquid SD/-Leu/-Trp medium for two days. A dilution series was made (1, 1/10 1, 1/100, 1/1000) and spotted on control plates (SD/-Leu/-Trp) and selective plates (SD/-Leu/-Trp/-His and SD/-Leu/-Trp/-His/-Ade, with or without 400 µM methionine to repress the bait expression). Yeast growth was recorded after incubation for 3 days at 30 °C.

## Statistics and reproducibility

The measurements in this study were conducted without any subjective bias. Phenotypic quantification, western blot quantification, and relative gene expression data were analyzed using either one-way or two-way analysis of variance (ANOVA) with Tukey's honest significant difference (HSD) in GraphPad Prism 8.0.1. In some cases, data were analyzed using Students' $t$ test with two-sample with parametric or unparametric distribution in GraphPad Prism 8.0.1 as well. For box plots, the box spans from the 25th percentile to the 75th percentile of the data. A horizontal line inside the box marks the median (50th percentile). Whiskers extend from the minimum to maximum. Regarding outliers, these were removed for further analyses in Appendix Figs. S15 and S21.

## Data availability

All data are available from the corresponding author upon request. Source data are provided with this paper. Python code that produces the model simulation results is available at https://gitlab.com/leahband/auxin_gravitropism_SLOMO.

The source data of this paper are collected in the following database record: biostudies:S-SCDT-10_1038-S44318-026-00746-8.

## Peer review information

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

## Acknowledgements

We thank Ke Xu for assistance with gravitropism and lateral root phenotyping and Bert De Rybel for lateral root cap/epidermis–specific promoter entry clones. LP, SZ, and TZ were supported by the Chinese Scholarship Council (201806870020, 201506230173, 202208410058) and LP also received support from the Ghent University Special Research Fund (01CD0523). MMG was supported by the Swiss National Funds (310030_197563). LRB's work was supported by the Biotechnology and Biological Sciences Research Council (grant number BB/M019837/1).

## Author contributions

**Lixia Pan**: Resources; Formal analysis; Funding acquisition; Validation; Investigation; Visualization; Methodology; Writing—original draft; Writing—review and editing. **Shanshuo Zhu**: Resources; Formal analysis; Funding acquisition; Validation; Investigation; Visualization; Methodology; Writing—review and editing. **Shao-Li Yang**: Formal analysis; Visualization; Methodology; Writing—review and editing. **Nathan Mellor**: Software; Formal analysis; Visualization. **Francesca R Iacobini**: Formal analysis. **Tingyu Zhu**: Formal analysis; Funding acquisition. **Pia Neyt**: Formal analysis. **Brigitte van de Cotte**: Formal analysis. **Michaël Vandorpe**: Formal analysis. **Ranjan Swarup**: Resources; Writing—review and editing. **Daniël Van Damme**: Supervision; Writing—review and editing. **Markus M Geisler**: Supervision; Funding acquisition; Writing—review and editing. **Kris Gevaert**: Supervision; Writing—review and editing. **Leah R Band**: Software; Supervision; Funding acquisition; Writing—review and editing. **Ive De Smet**: Conceptualization; Supervision; Funding acquisition; Writing—original draft; Project administration; Writing—review and editing.

Source data underlying figure panels in this paper may have individual authorship assigned. Where available, figure panel/source data authorship is listed in the following database record: biostudies:S-SCDT-10_1038-S44318-026-00746-8.

## Disclosure and competing interests statement

The authors declare no competing interests.

