## [Peer Review File · The EMBO Journal]

Dual ubiquitin signalling by SLOMO controls AUX1 activity and turnover during root gravitropism

Lixia Pan, Shanshuo Zhu, Shao-Li Yang, Nathan Mellor, Francesca Iacobini, Tingyu Zhu, Pia Neyt, Brigitte van de Cotte, Michael Vandorpe, Ranjan Swarup, Daniel Van Damme, Markus Geisler, Kris Gevaert, Leah Band, and Ive De Smet

Corresponding author(s): *Ive De Smet (ive.desmet@psb.vib-ugent.be)*

Review Timeline:

Submission Date:	8th Dec 25
Editorial Decision:	30th Jan 26
Revision Received:	5th Feb 26
Accepted:	25th Feb 26

Editor: *William Teale*

Transaction Report: The first round of review of this manuscript was performed at another journal.

Dear Ives,

As agreed, we sent your manuscript and the accompanying reviews to an independent expert in the field who acted in the role of arbitrating reviewer. We have now received their report which I have included below. As you will see, the reviewer concludes your manuscript is suitable for publication. However, before I can finally accept the manuscript, there are some editorial points which need to be addressed. In this regard would you please:

- remove the figures from the manuscript but keep the legends at the end,
- include the Code Availability statement in the Data Availability statement, and remove the separate title "Code Availability",
- rename the conflict of interest statement as the "Disclosure and competing interests statement",
- remove the AC/Credit section from the text,
- reformat references in alphabetical order, using 'et al.' after the tenth author in longer lists,
- remove the reference to 'data not shown' on page 9,
- complete and upload an author checklist (available to download in our submission guidelines),
- remove figures from the manuscript text and upload as individual production-quality Figure files (.tif, .eps, .pdf, or .jpg),
- provide a text callout for figure panel 4h,
- rename the supplementary information file as the Appendix the title page should have Appendix for "Title of the ms" and a Table of Contents with each item listed with its page number; the correct nomenclature for the items is Appendix Figure S1, etc. and Appendix Table S1 (the callouts in the manuscript also need updating); this file also has Suppl. Methods, please check if this needs to be moved to the manuscript file, if not, then the correct title should be Appendix Methods; this also applies to the References in the file (which should be Appendix References instead of Supplementary References),
- include a Reagents and Tools table,
- rename the Materials and Methods section as 'Methods',
- move the Acknowledgments and Disclosure statements, placing them after the Data Availability Statement,
- state p values in the legends of figures 3F, H; 4E, 5C,
- define box plots in terms of minima, maxima, centre, bounds of box and whiskers, and percentile in the legends of figures 3D, 5B, C, and
- define error bars in the legend of figure 2D.

We include a synopsis of the paper (see <http://emboj.embopress.org/>). Please provide me with a general summary image, a two sentence statement and 3-5 bullet points that capture the key findings of the paper.

I am looking forward to receiving your revised manuscript.

EMBO Press is an editorially independent publishing platform for the development of EMBO scientific publications.

Best wishes,

William

William Teale, PhD
Editor
The EMBO Journal
w.teale@embojournal.org

Read our guidance for manuscript revisions and related editorial policies: <https://link.springer.com/journal/44318/submission-guidelines#cms-Revised-submissions>

<https://media.springernature.com/original/springer-cms/rest/v1/content/27825798/data/v1>

- a point-by-point response to the referees' comments, with a detailed description of the changes made (as a word file).
- a word file of the manuscript text.

- individual production quality figure files (one file per figure)
- a complete author checklist
- Expanded View files (replacing Supplementary Information)
- a Reagents and Tools Table as part of the Methods section

Please remember: Digital image enhancement is acceptable practice, as long as it accurately represents the original data and conforms to community standards. If a figure has been subjected to significant electronic manipulation, this must be noted in the figure legend or in the 'Methods' section. The editors reserve the right to request original versions of figures and the original images that were used to assemble the figure.

We realize that it is difficult to revise to a specific deadline. In the interest of protecting the conceptual advance provided by the work, we recommend a revision within 3 months (30th Apr 2026). Please discuss the revision progress ahead of this time with the editor if you require more time to complete the revisions. Use the link below to submit your revision:

Referee #1:

The revised manuscript presents a convincing framework in which SLOMO-dependent ubiquitination targets functionally important residues in AUX1 and thereby modulates auxin transport during gravitropism. I feel the work is of high quality and that the reviewer 3 is quite critical. He/she is however also correct in pointing out that the data do not allow a strict causal separation between effects of ubiquitination and intrinsic roles of the modified lysines on transporter conformation. This limitation is now clearly acknowledged and appropriately discussed. In my view, this transparency strengthens rather than weakens the manuscript, and feel the work is suitable for publication in EMBO Journal.

All minor editorial requests have been addressed by the authors.

Dear Ives,

I am pleased to inform you that your manuscript has been accepted for publication in the EMBO Journal.

Congratulations to you and all involved!

You may qualify for financial assistance for your publication charges - either via a Springer Nature fully open access agreement or an EMBO initiative. Check your eligibility: <https://link.springer.com/journal/44318/how-to-publish-with-us>

Best wishes,

William

William Teale, PhD
Editor
The EMBO Journal
w.teale@embojournal.org

Please note that it is The EMBO Journal policy for the transcript of the editorial process (containing referee reports and your response letters) to be published as an online supplement to each paper. If you should prefer removal of any referee-only figures included in the point-by-point response(s), e.g. because they may still be used for future publication or because they have been reproduced from published work by others, please do let us know immediately via response email.
More information is available here: <https://link.springer.com/partners/embo-press/editorial-policies#Peer%20review>